# Metabolic landscape of the tumor microenvironment at single cell resolution

Zhengtao Xiao [1], Ziwei Dai [1] & Jason W. Locasale [1]

The tumor milieu consists of numerous cell types each existing in a different environment. However, a characterization of metabolic heterogeneity at single-cell resolution is not established. Here, we develop a computational pipeline to study metabolic programs in single cells. In two representative human cancers, melanoma and head and neck, we apply this algorithm to define the intratumor metabolic landscape. We report an overall discordance between analyses of single cells and those of bulk tumors with higher metabolic activity in malignant cells than previously appreciated. Variation in mitochondrial programs is found to be the major contributor to metabolic heterogeneity. Surprisingly, the expression of both glycolytic and mitochondrial programs strongly correlates with hypoxia in all cell types. Immune and stromal cells could also be distinguished by their metabolic features. Taken together this analysis establishes a computational framework for characterizing metabolism using single cell expression data and defines principles of the tumor microenvironment.

[1] Department of Pharmacology and Cancer Biology, Duke University School of Medicine, Durham, NC 27710, USA. Correspondence and requests for materials should be addressed to Z.D. (email: ziwei.dai@duke.edu) or to J.W.L. (email: jason.locasale@duke.edu)

Metabolic reprogramming in cancer cells supports specific demands for energy, biomass, redox maintenance, and cellular communication[1]. Cellular metabolism is shaped by both genetic and environmental factors including the somatic driver mutations selected during tumor evolution, the tissue of origin, and the local nutrient environment[1–5]. Given the importance of environmental factors, tumor metabolism exhibits grossly different properties in laboratory cell culture settings than in vivo. Differences in the metabolism of humans and model organisms are also substantial. Nevertheless, most conclusions surrounding tumor metabolism have been obtained using cell culture models and model organisms and the number of direct observations of cellular metabolism are few. Indeed, nearly all observations of human tumor metabolism in vivo have been conducted using measurements obtained from bulk tumors. These findings have advanced our understanding of tumor metabolism tremendously. Nevertheless, they do result from a population average over the genetic and environmental variables of each cell.

There are numerous sources of intratumoral heterogeneity. Tumor cells exist within a microenvironment consisting of stromal cells such as cancer-associated fibroblasts (CAFs), immune cells, endothelial cells and many others. Each of these cell types takes an active role in tumor cell proliferation. For example, CAFs may release cytokines and growth factors that are received by and function to signal in cancer cells[6]. The immune compartment of an established malignant tumor is collectively immunosuppressive[7]. Endothelial cells provide vasculature to provide nutritional support in challenging environments[8]. Each cell type has unique metabolic demands that enable specific function. In addition to the unique metabolic demands of each cell type, each cancer experiences a distinct nutrient environment, distinct engagement of extracellular signals, and may derive from a different cell of origin thus possibly having distinct mutational patterns[9]. Therefore at the cellular level, each cell within the tumor is likely to have a different metabolic status[10–13]. Nevertheless, direct observations of cellular metabolism in vivo at the single-cell level is difficult. Most conclusions about the nature of the tumor microenvironment have relied on in vitro models such as co-culture systems[14–16] or measurements of single variables such as immunohistochemical staining for the expression of a metabolic enzyme[17].

Metabolism and its associated phenotypic biology are governed by the concentrations of metabolites and the rates or fluxes by which one metabolite is converted to another. A comprehensive understanding of metabolism requires knowledge of both concentrations and fluxes. These measurements are difficult to obtain in humans and have so far been exclusively conducted in bulk tumors. Global gene expression however is readily measurable and the advent of single-cell sequencing technologies enables expression profiling of individual cells within entire tissues or tumors[18–23]. It is also an indirect means of assessing metabolism. Nevertheless, gene expression has provided many insights into metabolic pathway activity and in many documented instances the gene expression is predictive of metabolic flux[24,25]. Thus single-cell sequencing could provide some insight into metabolism at the single-cell level in human tumors.

In this study, we analyze metabolic gene expression profiles of more than 9000 single cells from two representative human tumor types including melanoma[20] and squamous cell carcinoma of the head and neck (HNSCC)[22]. We find that activities of metabolic pathways in malignant cells are in general more active and plastic than those in non-malignant cells, and the metabolic features of single cancer cells are poorly captured by measurements done with bulk tumors. Variation in mitochondrial activity is the major contributor to the metabolic heterogeneities among both malignant cells and non-malignant cells, and, strikingly, the activities of glycolysis and oxidative phosphorylation both correlate with hypoxia at the single-cell level. We also identify metabolic features of different immune and stromal cell subtypes and find patterns distinct from behaviors of these cells in ex vivo culture conditions. These findings begin to unravel principles of how malignant transformation affects the metabolic phenotypes of tumor and non-tumor cells within the tumor microenvironment.

## Results

**Landscape of metabolic gene expression at single-cell level**. We developed a computational pipeline for analyzing metabolic gene expression profiles at the single-cell level (Fig. 1a, Methods). In brief, we applied missing data imputation and data normalization to gene expression profiles to account for the influence of technical noise. We then characterized the global structure of single-cell metabolic programs using clustering analysis, identified cell type-specific metabolic programs using quantitative metrics we developed, and designed algorithms for quantitation of metabolic heterogeneity of malignant and non-malignant cells. We applied this pipeline to two single-cell RNA-seq (scRNA-seq) datasets for human melanoma[20] and HNSCC[22], which include an expansive set of gene expression of 4054 cells and 5502 cells respectively (Methods). These datasets were selected because they covered the largest numbers of cells and included detailed annotation of the cell types, while many of the currently available scRNA-seq datasets for human cancer are limited by cell number and sequencing depth. Each dataset covers both malignant and non-malignant cells isolated from patient-derived human tumors with different genotypic and phenotypic backgrounds (Fig. 1b, c), thus enabling an in-depth investigation of the expression of metabolic genes and pathways in each cell type.

We first analyzed the global structure of metabolic gene expression in these two datasets using t-distributed stochastic neighbor embedding (t-SNE)[26] based on expression levels of 1566 metabolic genes (Methods). Imputation of zero values was performed before applying t-SNE to correct for the influence of the high frequency of dropout events (i.e. failure in detecting expressed genes due to low sequencing depth) in certain cell types (Methods). The imputation effectively reduced the dropout rates in all cell types (Supplementary Fig. 1a–d) without changing the pattern of the metabolic gene expression (Supplementary Fig. 1e, f). Clustering analysis after dimensionality reduction with t-SNE showed that metabolic gene expression profiles of the malignant cells formed distinct clusters that corresponded to their tumors of origin (i.e. from which tumor the cell was derived) for both melanoma (Fig. 1d) and HNSCC (Fig. 1e), suggesting that metabolic gene expression in malignant cells is largely set by patient-specific factors. This was further corroborated by the higher correlation coefficients of metabolic gene expression profiles between malignant cells from tumors within the same patient (average Spearman's correlation = 0.91) than those between cells from tumors from different patients (average Spearman's correlation = 0.79, Fig. 1f, g). Cells from patients of the same genotypic background also showed higher similarity than those from different genotypic backgrounds (average Spearman's correlation = 0.87 compared to 0.79 for melanoma, 0.87 compared to 0.77 for HNSCC, Fig. 1f, g). In contrast, metabolic gene expression profiles of non-malignant cells showed no distinguishable differences between patients (average difference between intratumoral and intertumoral Spearman's correlation = 0.01 for non-malignant cells in melanoma and 0.007 for those in HNSCC, Fig. 1h, i), indicating that metabolism of these normal cells in the tumor microenvironment exhibits no

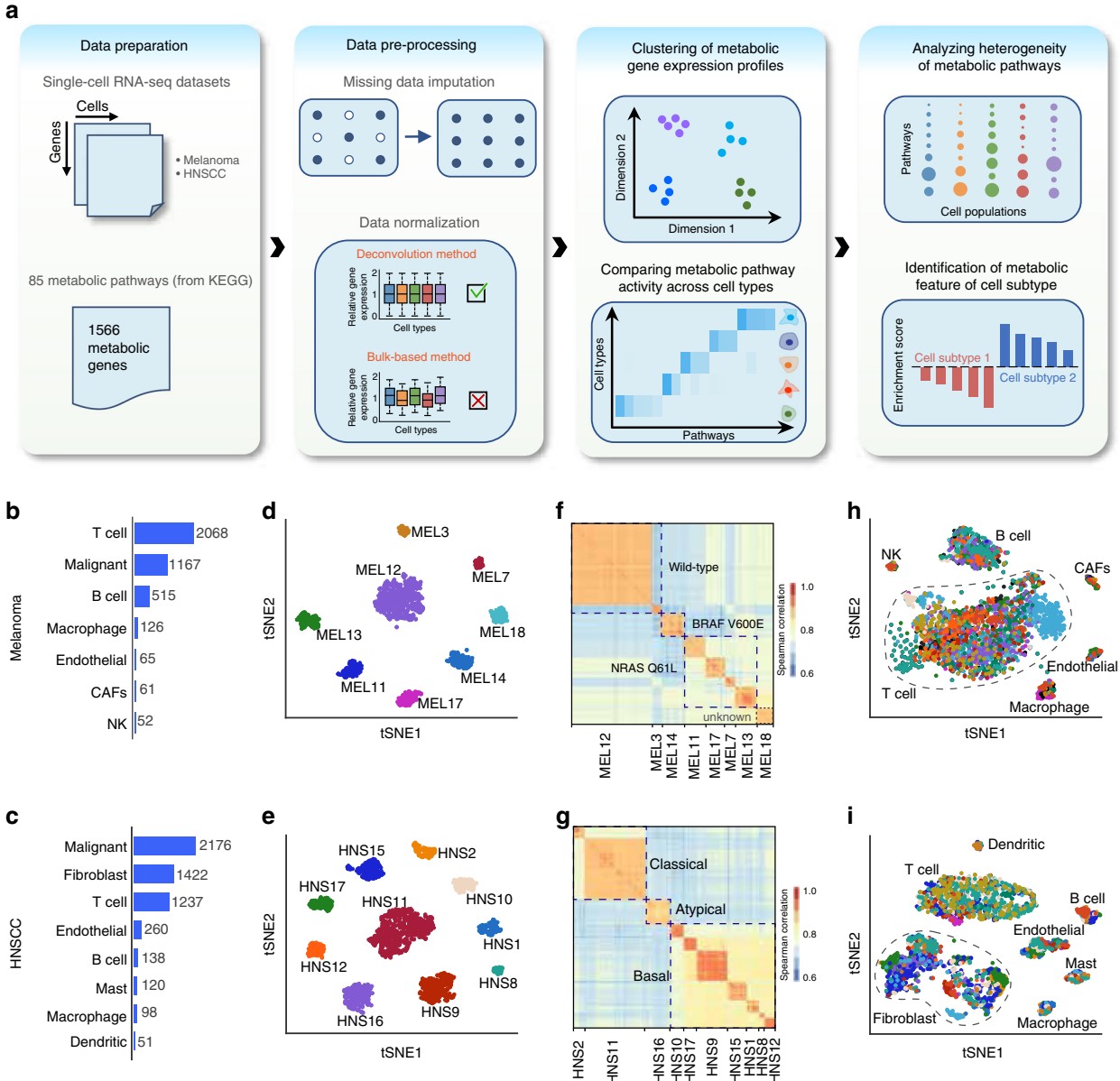

**Fig. 1** Landscape of metabolic gene expression at single-cell level. **a** Schematic representation of the scRNA-seq data analysis pipeline. **b** Numbers of each type of cells in the melanoma dataset. **c** Numbers of each type of cells in the head and neck squamous cell carcinoma (HNSCC) dataset. **d** t-SNE plot of metabolic gene expression profiles of malignant cells from the melanoma dataset. The color of each dot indicates the tumor which the cell comes from. **e** Same as in **d** but for the HNSCC dataset. **f** Clustered correlation matrix showing Spearman's rank correlation coefficients of metabolic gene expression profiles between malignant cells in the melanoma dataset. **g** Same as in **f** but for the HNSCC dataset. **h** t-SNE plot of metabolic gene expression profiles of non-malignant cells from the melanoma dataset. The color of each dot indicates the tumor which the cell comes from. **i** Same as in **h** but for the HNSCC dataset

observable interpatient heterogeneity. Similar trends were also observed for the distributions of correlation coefficients of metabolic gene expression between cells from the same patient or different patients, in which only malignant cells showed significantly stronger intratumoral correlation than intertumoral correlation (Supplementary Fig. 2). We next repeated the t-SNE analysis on a randomly selected set of genes (Supplementary Fig. 3a–d) and the complete set of genes (Supplementary Fig. 3e–h) and found that the clustering patterns of these gene sets were similar (relative mutual information close to 1, Supplementary Table 1, Supplementary Methods) to that of the metabolic genes, implying that the metabolic plasticity of malignant cells reflects their intrinsic flexibility in gene expression program that influences both metabolic and non-metabolic genes.

Taken together, these results suggest that malignant cells exhibit higher metabolic plasticity which likely leads to patient-specific metabolic reprogramming of cancer cells but not the supporting cells in the tumor microenvironment.

**Cell type-specific metabolic reprogramming.** We next sought to identify the overall features of metabolic pathway variation among the different cell types, especially between malignant and non-malignant cells. To quantify the activity of a metabolic pathway, we developed a pathway activity score defined as the relative gene expression value averaged over all genes in this pathway and all cells of this type (Fig. 2a, b, Methods). To enable comparison between different cell types, we normalized the gene

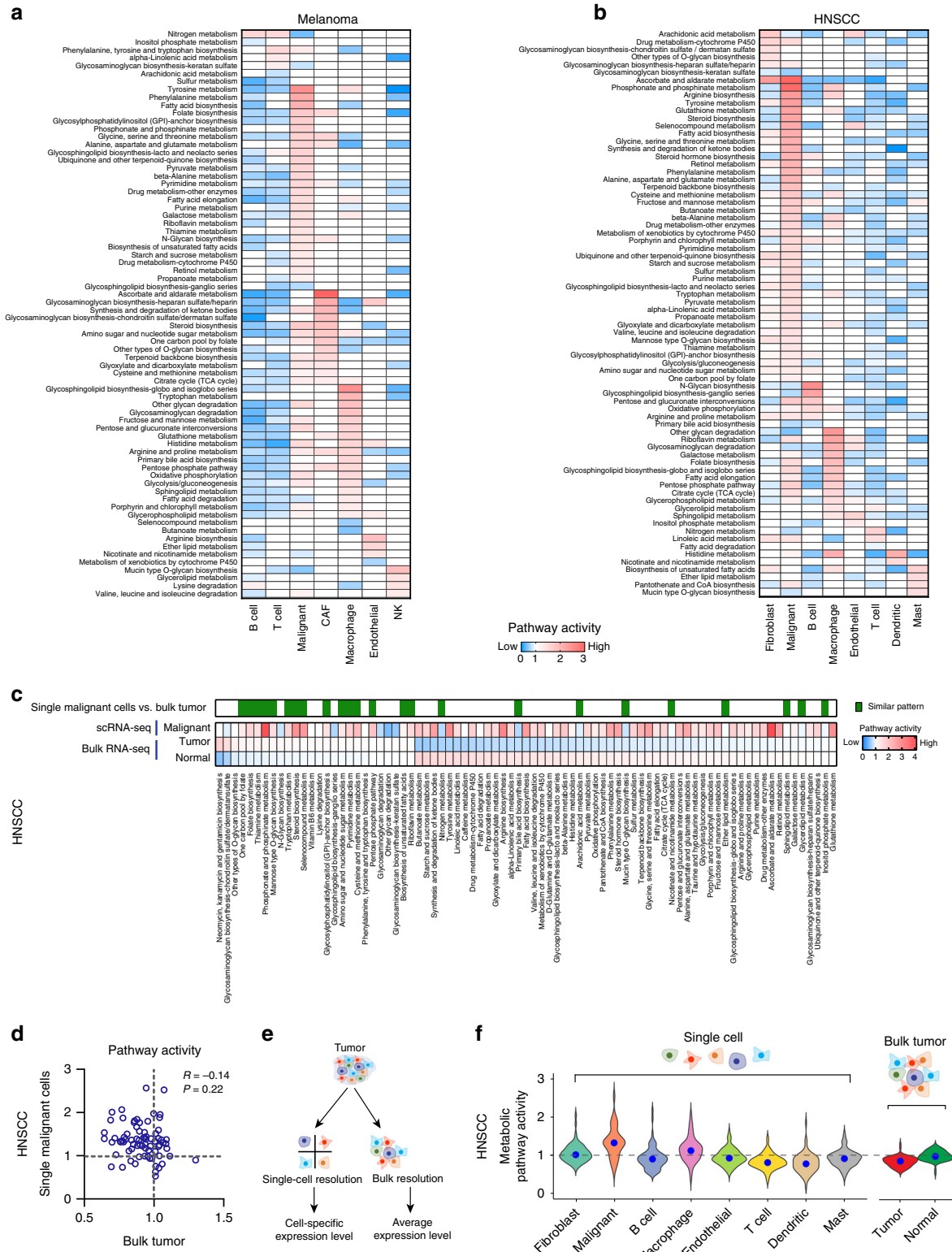

**Fig. 2** Cell type-specific metabolic reprogramming. **a** Metabolic pathway activities in cell types in the melanoma dataset. Statistically non-significant values (random permutation test p > 0.05) are shown as blank. **b** Metabolic pathway activities in cell types in the HNSCC dataset. Statistically non-significant values (random permutation test p > 0.05) are shown as blank. **c** Metabolic pathway activities in HNSCC tumor samples and matched adjacent normal samples from TCGA computed based on bulk RNA-seq data. The color bar on the top marks the pathways with similar activity changes in single malignant cells compared to single non-malignant cells and bulk tumors compared to normal tissue samples. **d** Scatter plot comparing pathway activities between bulk HNSCC tumors in TCGA and single malignant cells in the HNSCC scRNA-seq dataset. **e** Difference between bulk and single-cell RNA-seq in characterizing gene expression profiles in tumors. **f** Distributions of pathway activities in different cell types from the HNSCC scRNA-seq dataset (left) and in bulk tumors and normal samples from TCGA (right)

expression values using a deconvolution method[27] which resulted in the highest similarity of normalized expression value distributions between cell types among four commonly used data normalization methods (Supplementary Fig. 4, Methods). Among the 80 metabolic pathways with at least 5 genes included, over 70 were more highly expressed (pathway activity score >1 and permutation test $p$-value < 0.01) in at least one cell type compared to other cell types, indicating that the activities of most metabolic pathways are determined by cell type. As an example, the differential expression of oxidative phosphorylation (OXPHOS) and glycolysis across cell types was further supported by directly comparing distributions of the mean expression level of genes within these pathways across different cell types (one-way ANOVA $p$-value < 0.01 for both datasets, Supplementary Fig. 5). To further characterize the cell type-specific metabolic features, we grouped the metabolic pathways into 11 categories based on KEGG classifications (Supplementary Fig. 6a, b), and assessed enrichment of each category in pathways up-regulated in each cell type using a one-sided Fisher's exact test. These categories reflected different aspects of cellular metabolism such as carbohydrate metabolism, amino acid metabolism and nucleotide metabolism. To our surprise, most categories (11 out of 11 for melanoma and 10 out of 11 for HNSCC) were not enriched in pathways up-regulated in a specific cell type (i.e. one-sided Fisher's exact $p$-value > 0.05 for all cell types, Supplementary Fig. 6c, d), indicating that each cell type undergoes global up- or down-regulation of metabolic pathways in all categories. Malignant cells had the largest number of metabolic pathways significantly up-regulated in both melanoma (45 pathways up-regulated in malignant cells compared to 20 in cancer-associated fibroblasts, 22 in macrophages and <10 in all other cell types, Fig. 2a, Supplementary Data 1) and HNSCC (56 pathways up-regulated in malignant cells compared to 28 in fibroblasts, 24 in macrophages, 11 in B cells and <10 in all other cell types, Fig. 2b, Supplementary Data 1), and the up-regulated pathways included many different parts of cellular metabolism such as central carbon metabolism, one carbon metabolism, methionine metabolism, steroid biosynthesis, and beta-alanine metabolism. Thus, compared to non-malignant cells, malignant cells undergo a global up-regulation of metabolic activity. Importantly, the global up-regulation was only observed for metabolic pathways but not for non-metabolic pathways (median value of non-metabolic pathway activities close to 1 for all cell types, Supplementary Fig. 6e–g). Interestingly, comparison of pathway activities between melanoma and HNSCC in the cell types shared by the two datasets (Supplementary Fig. 6h) revealed high concordance between the two tumor types in the metabolic features of T cells (Spearman's correlation = 0.58, $p$-value = 3.6e-8), malignant cells (Spearman's correlation = 0.45, $p$-value = 4.9e-5), endothelial cells (Spearman's correlation = 0.43, $p$-value = 1.2e-4) and macrophages (Spearman's correlation = 0.44, $p$-value = 6.9e-5), while metabolic pathway activities of B cells and fibroblasts correlated poorly between the two tumor types (Spearman's correlation < 0.1, $p$-value > 0.01 for both cell types), suggesting that metabolism of these cell types is more sensitive to environmental factors and the effector status of these immune cells.

To evaluate whether patterns of pathway activities in single malignant cells were consistent with the more commonly used transcriptomic profiling of bulk tissue samples, we computed pathway activity scores based on RNA-seq data for bulk tumor samples and matched normal tissue samples from The Cancer Genome Atlas (TCGA)[28] and compared the results to our analysis in single malignant cells. Since there are no matched normal samples for melanoma in the TCGA database, here we only considered the HNSCC dataset which contains 43 paired tumor and normal samples[29]. First, to confirm that the gene

expression values in TCGA bulk tumors are comparable to tumors in the single-cell RNA-seq dataset, we reconstructed bulk gene expression profiles for tumors in the single-cell dataset by pooling gene expression profiles of all single cells derived from one tumor. We then compared them to those in the TCGA database to show that the reconstructed bulk tumors, but not the negative control that we generated by pooling 500 single T cells, showed a very strong correlation of gene expression levels with the TCGA tumor samples (average Pearson's $R = 0.77$ for reconstructed bulk tumors compared to 0.69 for pooled T cells, Supplementary Fig. 7), suggesting that tumor samples in the single-cell dataset and those in TCGA are approximately equivalent. We found 6 pathways up-regulated and 34 pathways down-regulated in tumor samples compared to normal samples (Permutation test $p$-value < 0.01, Fig. 2c). Among these pathways, only 24 showed a consistent pattern of activity in single malignant cells and bulk tumors (Fig. 2c). Notably, 25 out of the 56 pathways up-regulated in the single malignant cells were identified as down-regulated in tumors based on the bulk RNA-seq, and the pathway activities correlated poorly (Pearson's $R = -0.14$, $p$-value = 0.22) between bulk tumors and single malignant cells (Fig. 2d). The discrepancy between single-cell and bulk RNA-seq in identifying tumor-associated metabolic pathway activities is likely due to the intrinsic heterogeneity in cellular composition of the tumors (Fig. 2e). Bulk RNA-seq measures the average expression levels over a mixture of different cell types thus masking the difference between cell types in the same sample. This was further supported by an analysis of the distributions of pathway activities in single cells and bulk samples showing higher variation of pathway activities between different types of single cells than between bulk tumors and normal tissues (average standard deviation of pathway activities = 0.27 for single cells compared to 0.14 for bulk samples, Fig. 2f). The higher metabolic activity and variation in single malignant cells compared to bulk tumors was further confirmed by comparison of metabolic pathway activities between single malignant cells and reconstructed bulk tumors (one-sided Wilcoxon's rank-sum test $p$-value = 2.5e-5 for melanoma and 4.6e-4 for HNSCC, Kolmogorov-Smirnov test $p$-value = 2.9e-5 for melanoma and 8.7e-5 for HNSCC, Supplementary Fig 8). Taken together, these results reveal a global enhancement of metabolic activity in single malignant cells which can only be detected with gene expression profiling at the single-cell level.

**Intratumoral metabolic heterogeneity of malignant cells.** In addition to the common routes of metabolic reprogramming in malignant cells, metabolism is also affected by a location-specific, fluctuating nutrient supply and interactions with other neighboring cells in space. It is thus intriguing to investigate what parts of cellular metabolism are impacted by these environmental factors. To identify major contributors to intratumoral metabolic heterogeneity of malignant cells (i.e. variation of metabolism among malignant cells from the same tumor), we performed principal component analysis (PCA) and gene set enrichment analysis (GSEA)[30] to identify metabolic pathways enriched in genes explaining most of the variance among malignant cells in each tumor (Fig. 3a, Supplementary Fig. 9). We found that for both melanoma and HNSCC, OXPHOS was the top-scoring pathway in most tumors (Fig. 3b, c). Similarly, tricarboxylic acid cycle (TCA cycle) also showed a substantial contribution to metabolic heterogeneity in several tumors, indicating that variation in mitochondrial activity is the major contributor to intratumoral metabolic heterogeneity of malignant cells. We also tested two alternative metrics including coefficient of variation (CV, Supplementary Fig. 10a) and standard deviation (SD,

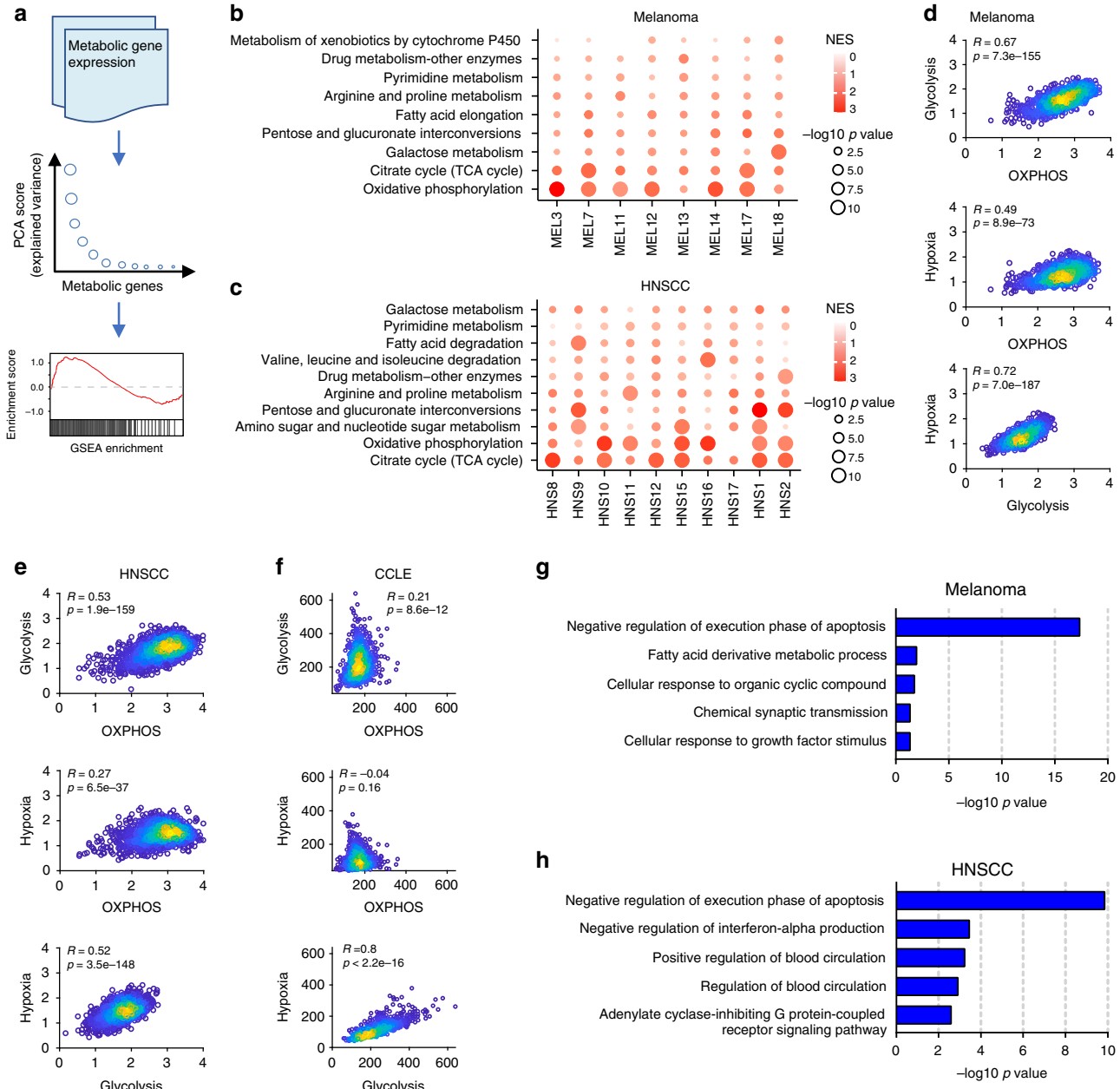

**Fig. 3** Intratumoral metabolic heterogeneity of malignant cells. **a** Workflow for quantitating metabolic heterogeneity of malignant cells. **b** Metabolic pathways enriched in genes with highest contribution to the metabolic heterogeneities among malignant cells from different tumors in the melanoma dataset. **c** Same as in **b** but for the HNSCC dataset. **d** Scatter plots comparing activities of glycolysis, OXPHOS and response to hypoxia in single malignant cells from the melanoma dataset. Colors of points indicate local density of points. **e** Same as in **d** but for the HNSCC dataset. **f** Same as in **d** but for cancer cell lines from CCLE. **g** GO terms enriched in genes up-regulated in cells with lowest expression levels of glycolysis, OXPHOS and hypoxia pathways in the melanoma dataset. **h** Same as in **g** but for the HNSCC dataset

Supplementary Fig. 10b) on the melanoma dataset to exclude potential bias in choice of metric. We found that the top-scoring pathways identified by SD (Supplementary Fig. 10b) were highly consistent with those by computed PCA score (Fig. 3b), and PCA score and SD were less dependent on average gene expression levels compared to the CV which tends to be higher for low abundant genes (Supplementary Fig. 10c-e). Moreover, the top-scoring pathway identified by CV, steroid hormone biosynthesis, showed visibly less variation in expression and much lower expression levels compared to OXPHOS (Supplementary Fig. 10f).

We next sought to investigate the coupling between mitochondrial activity and environmental factors such as oxygen and other nutritional supplies in single malignant cells. Since direct measurements of nutritional status of the cells are not available, we used the average expression level of a set of genes known to respond to hypoxia as a metric of oxygen supply which is an environmental factor known to have great impact on cellular metabolism (Methods). We used the average expression level of genes in OXPHOS as a surrogate of mitochondrial activity. We also considered glycolysis since it is another important pathway in supplying energy and material for cell survival and proliferation, and its relationship with OXPHOS in cancer metabolism is still a matter of interest. We found that activity of glycolysis and the hypoxia signature were highly correlated in both melanoma and HNSCC (Pearson's $R = 0.72$ for melanoma and 0.52 for

HNSCC, Fig. 3d, e), which agrees with previous studies showing that hypoxia increases glycolytic activity[31,32]. To our surprise, we also found that OXPHOS significantly correlated with glycolysis (Pearson's $R = 0.67$ for melanoma and 0.53 for HNSCC, Fig. 3d, e) and the response to hypoxia (Pearson's $R = 0.49$ for melanoma and 0.27 for HNSCC, Fig. 3d, e). To exclude the possibility that the positive correlations were driven by a few correlated genes with high variation in expression levels while most other genes were not correlated, we confirmed that most pairwise correlation coefficients of expression levels between individual genes from pathways were also positive (Supplementary Fig. 11). Notably, the correlations between OXPHOS and glycolysis and that between OXPHOS and hypoxia were lost in cultured cancer cells from the Cancer Cell Line Encyclopedia (CCLE)[33] (Fig. 3f), suggesting that the unexpected coupling between aerobic respiration and hypoxia-related pathways is an unique feature of single malignant cells in the tumor microenvironment. Moreover, for both melanoma and HNSCC, malignant cells with lowest expression levels of OXPHOS, glycolysis and hypoxia were associated with a gene expression feature that up-regulates genes related to the GO term negative regulation of execution phase (Fig. 3g, h, Supplementary Fig. 12), suggesting that this subpopulation of cells exhibited reduced apoptosis which may facilitate cancer progression. These results together indicate that mitochondrial OXPHOS is the most important contributor to intratumoral metabolic heterogeneity, and OXPHOS is not mutually exclusive with glycolysis as routes for energy production in adapting to the tumor microenvironment with varying oxygen supply.

**Metabolic heterogeneity of non-malignant cells**. We next explored the metabolic heterogeneity of non-malignant cells. Since metabolic gene expression profiles of non-malignant cells were clustered according to cell type, and there were no significant differences between different tumors for the same cell type (Fig. 1h, i and Supplementary Fig. 2a, b), We focused on identifying the major contributors to metabolic heterogeneity in each non-malignant cell type. For each non-malignant cell type, we repeated the PCA and GSEA analyses as we did for malignant cells and found that variations in OXPHOS also dominated the metabolic heterogeneity in all non-malignant cell types (Fig. 4a, b, Supplementary Fig. 13). In addition to OXPHOS, TCA cycle also substantially contributed to the metabolic heterogeneity of all non-malignant cell types. These results demonstrate that similar to the case of malignant cells, mitochondrial activity is also the major contributor to metabolic heterogeneity in non-malignant cells.

To evaluate how mitochondrial activity in non-malignant cells relates to oxygen availability, we correlated activities of OXPHOS and glycolysis with the hypoxia-associated feature in each cell type (Fig. 4c, d). Similar to the case of malignant cells, glycolysis, OXPHOS and the hypoxia feature were also significantly correlated in almost all cell types except for macrophages in the HNSCC dataset, in which OXPHOS and hypoxia showed no significant correlation (Pearson's $R = -0.02$, $p$-value = 0.87). These findings, together with the results for malignant cells, challenge the long-standing concept that metabolic reprogramming of central carbon metabolism frequently takes the form of a switch between glycolysis and mitochondrial respiration. At the single-cell level, it is plausible that cells facing more severe oxygen deprivation tend to up-regulate both glycolysis and mitochondrial OXPHOS, which may help cells to more effectively compete with other cells for limited resources. Such a positive correlation between the hypoxia signature and energy-producing pathways is conserved in almost all cell types included in the single-cell RNA-seq datasets.

**Metabolic features of non-malignant cell subtypes**. Non-malignant cells such as immune and stromal cells are important constituents of the tumor microenvironment. These cells are known to differentiate into subtypes with distinct roles, and this process involves metabolic reprogramming to satisfy their cell autonomous metabolic demands and enable interactions with other cell types[10,11,34,35]. Next, we use scRNA-seq to characterize the metabolic features of T cell and fibroblast subpopulations which together constitute the largest non-malignant cell populations in the melanoma and HNSCC datasets. T cells were first separated into CD8+ and CD4+ subtypes based on the expression of *CD4* and *CD8A*[36] (Fig. 5a, Methods). The CD4+ T cells were further classified into regulatory T cells (Tregs) and T helper cells (Ths) based on expression levels of *FOXP3* and *CD25* which are known to be specifically expressed in these specific cell types[37] (Fig. 5a, Methods). We then performed GSEA analysis to identify metabolic pathways enriched in each subtype. We found that OXPHOS was the most important metabolic pathway distinguishing T cell subtypes: CD4+ T cells exhibited significantly higher levels of OXPHOS compared to CD8+ T cells in both melanoma (GSEA $p$-value = 0.002) and HNSCC (GSEA $p$-value < 0.001, Fig. 5b, c, Supplementary Fig. 14a, b, Supplementary Data 2). Interestingly, compared to Ths, Tregs exhibited up-regulation of glycolysis (GSEA $p$-value < 0.001 for both tumors) in addition to OXPHOS (GSEA $p$-value < 0.001 for both tumors, Fig. 5d, e, Supplementary Fig. 14c, d, Supplementary Data 2). This appears to contradict with previous studies showing that among immune cells derived from healthy mice not bearing tumors, Ths tend to be more glycolytic compared to Tregs[38]. On the other hand, the OXPHOS preference of CD4+ T cells and Tregs is consistent with previous studies[39–41], highlighting enhanced mitochondrial oxidative metabolism as a universal metabolic feature of these T cell subtypes in different contexts. These results suggest that subpopulations of immune cells in the tumor microenvironment have metabolic features that differ from their behaviors in normal tissues.

We next characterized the metabolic features of subpopulations of fibroblasts which serve as a major component of tumor stroma and have diverse roles in both normal functions such as wound healing and tumor-promoting functions such as remodeling the extracellular matrix and interacting with tumor cells to support their growth[42]. According to a previous study[22], the 1422 fibroblasts in the HNSCC dataset formed two major subpopulations exhibiting gene expression corresponding to CAFs or myofibroblasts, respectively (Fig. 5f, Methods). We thus performed GSEA analysis to compare metabolic gene expression between the two fibroblast subtypes (Fig. 5g, Supplementary Fig. 14e, Supplementary Data 2). We found significant up-regulation of 15 metabolic pathways (GSEA $p$-value < 0.05) that distinguished CAFs from myofibroblasts. On the other hand, inositol phosphate metabolism was the only metabolic pathway up-regulated in myofibroblasts ($p$-value < 0.001), indicating that CAFs are more metabolically active compared to myofibroblasts. Glycolysis was significantly up-regulated in CAFs (GSEA $p$-value = 0.048), which is in line with a hypothesized metabolic feature of CAFs, in which CAFs exhibit enhanced glycolysis which produces excess lactate that can be utilized by adjacent tumor cells to support growth[43,44]. Notably, we also found several other groups of metabolic pathways up-regulated in CAFs. These pathways included arachidonic acid metabolism (GSEA $p$-value < 0.001) and linoleic acid metabolism (GSEA $p$-value = 0.002), which are known to produce inflammatory mediators[45], and a group of pathways related to glycan biosynthesis and degradation. Up-regulation of these pathways in CAFs may support the function of CAFs in secreting small molecule compounds and proteins to remodel the tumor microenvironment. Taken together, these results identify a

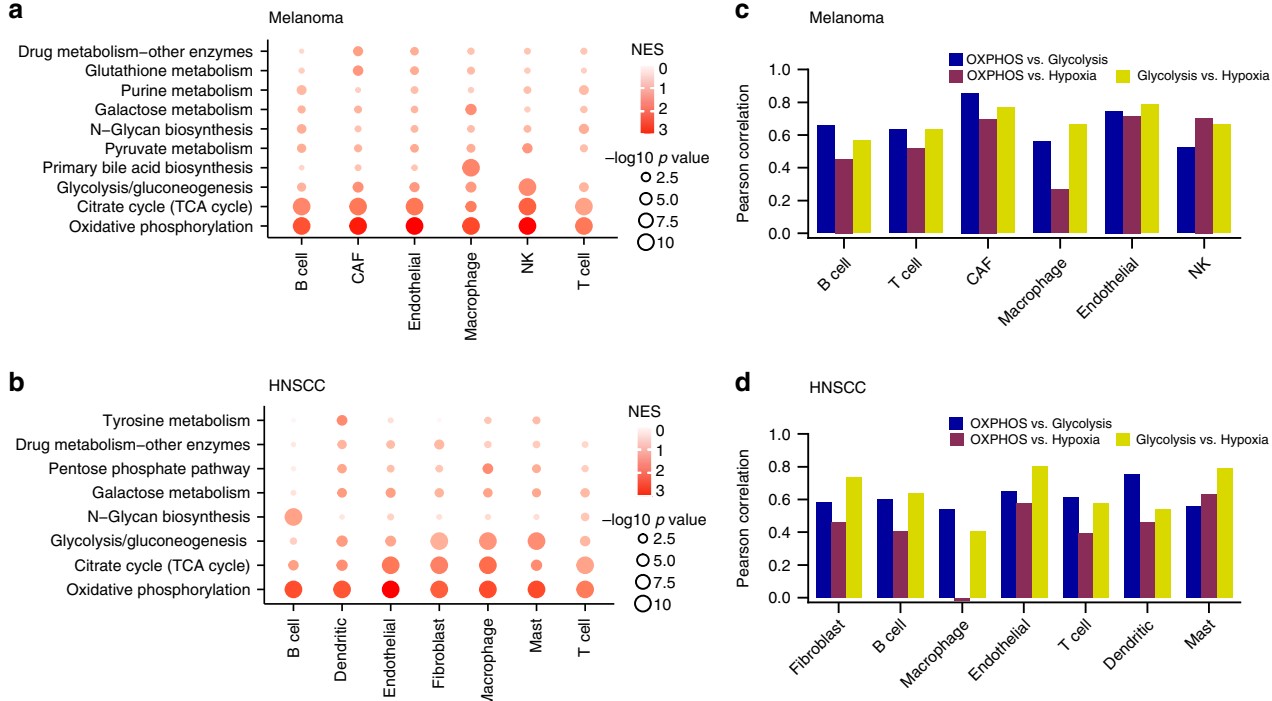

**Fig. 4** Metabolic heterogeneity of non-malignant cells. **a** Metabolic pathways enriched in genes with highest contribution to the metabolic heterogeneities among different types of non-malignant cells from the melanoma dataset. **b** Pearson's correlation coefficients between the activities of glycolysis, OXPHOS and response to hypoxia in non-malignant cells from the melanoma dataset. **c** Same as in **a** but for the HNSCC dataset. **d** Same as in **b** but for the HNSCC dataset

metabolic phenotype of CAFs that potentially helps establish their roles in interacting with other cell types and modulating the tumor microenvironment.

## Discussion
We have characterized the rewiring of metabolic pathways in single malignant cells compared to their normal partners in the same tumor microenvironment by analyzing metabolic gene expression profiles of single malignant and non-malignant cells. Although metabolic gene expressional levels are not equivalent to metabolic fluxes or metabolite abundance, there is some evidence that metabolic gene expression can be to some if not a large extent predictive of metabolic fluxes, and metabolite concentrations[24,46]. Thus, our findings with the single-cell gene expression profiles provide clues about the overall trends of metabolic activities in single cells. We found that, compared to non-malignant cells, malignant cells not only exhibit high metabolic plasticity that allows them to adapt their metabolism to different genotypic and environmental contexts, but also follow a common pattern of global up-regulation of activities of metabolic pathways in almost all functional categories. The global up-regulation of metabolic genes but not non-metabolic genes implies that malignant cells allocate more transcriptional resources to the expression of metabolic genes and having likely higher fluxes for most metabolic reactions. These results point to the principle that metabolism of cancer cells is in general more flexible and active than that of non-malignant cells. Notably, most of the metabolic changes detected in single malignant cells compared to single non-malignant cells were not captured by comparing expression levels of metabolic genes between bulk tumor and normal samples, implying that comparison of metabolic configurations between tumors and normal tissues based on bulk measurements tends to underestimate the differences

between malignant and non-malignant cells due to the highly complicated cellular composition of the bulk samples. Consistently, our results appear to be different from previous studies comparing metabolic network expression in tumor and normal tissues[47,48].

There are several interesting findings around the role of mitochondrial activity in shaping the metabolic heterogeneity of tumors. First, variation in OXPHOS gene expression is the most important contributor to the metabolic heterogeneity among malignant cells from the same tumor and that among non-malignant cells of the same type. The high variation in OXPHOS activity suggests that this pathway might be responsible for adapting to environmental factors. It is thus interesting to investigate how such variability in OXPHOS activity contributes to tumor progression.

The role of mitochondria (OXPHOS and TCA cycle) in cancer is still a matter of debate. In addition to the well-known Warburg effect[49], several studies comparing metabolic gene expression between bulk tumors and normal tissues have also identified suppression of OXPHOS as a recurrent metabolic phenotype in tumors[47,50–52]. However, there are also numerous studies showing that active OXPHOS is in fact required for cancer progression. Mitochondrial inhibitors such as metformin are known to suppress cancer cell growth[53–55]. In this study, we found that OXPHOS gene expression levels were in general higher in single malignant cells (Fig. 2a, b, Supplementary Fig. 5), which appears to contradict observations based on bulk gene expression levels[47,50–52]. Further work is needed to resolve the discrepancy between single-cell and bulk RNA-seq in evaluating the role of OXPHOS in tumors, but it is likely due to the complexity of cellular composition of tumors that is almost impossible to be dissected by bulk measurements.

Another interesting finding about OXPHOS activity in single cells is that it is correlated with both glycolysis and response to

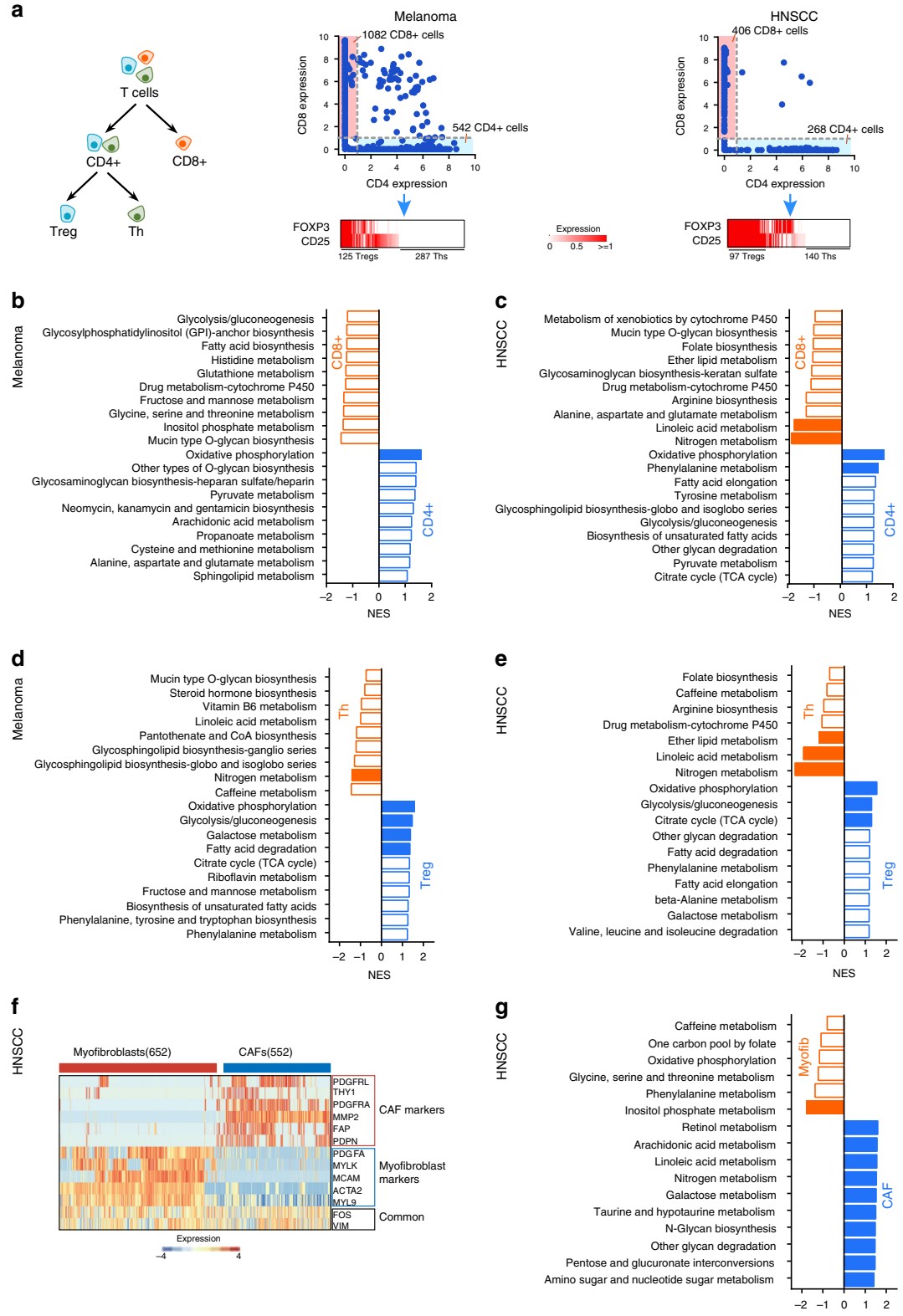

**Fig. 5** Metabolic features of non-malignant cell subtypes. **a** Left panel: classification of T cells into CD4$^+$, CD8$^+$, regulatory T cells (Tregs) and T helper cells (Ths). Middle and right: expression levels of the gene markers used for separating T cell subtypes in melanoma (middle) and HNSCC (right) datasets. **b** Top 10 metabolic pathways enriched in CD4$^+$ or CD8$^+$ T cells in the melanoma dataset. Significantly enriched pathways with GSEA $p$-value < 0.05 are highlighted in red (higher in CD8$^+$) or blue (higher in CD4$^+$). **c** Same as in **b** but for the HNSCC dataset. **d** Top 10 metabolic pathways enriched in Tregs or Ths in the melanoma dataset. Significantly enriched pathways with GSEA $p$-value < 0.05 are highlighted in red (higher in Th) or blue (higher in Treg). **e** Same as in **d** but for the HNSCC dataset. **f** Gene markers and their expression levels used for classifying fibroblast cells in the HNSCC dataset into CAFs and myofibroblasts. **g** Top 10 metabolic pathways enriched in CAFs or myofibroblasts in the HNSCC dataset. Significantly enriched pathways with GSEA $p$-value < 0.05 are highlighted in red (higher in myofibroblasts) or blue (higher in CAFs)

hypoxia in almost all cell types. This at the first glance is counterintuitive because hypoxia activates signal transduction pathways that induce glycolysis and suppress OXPHOS and other mitochondria-associated pathways[32]. Nevertheless, OXPHOS also has established role in mediating the response to hypoxia by serving as a sensor of oxygen availability through stabilization of hypoxia-induced factors (HIF)[56,57]. Therefore, the interplay between glycolysis, OXPHOS, and hypoxia is highly dynamic in living cells, and the quantitative relationship between them is at least partly determined by the interaction between the inhibitory effects of the HIF signaling pathway and the positive feedback from OXPHOS activity to HIF signaling in response to oxygen availability. Our analysis of single-cell transcriptomic profiles clarifies that activities of these pathways tend to be positively correlated in single cells from the tumor microenvironment with scarce and fluctuating oxygen supply. Whether such coupling at single-cell level exists in other types of tumor and whether it benefits the cancer cells needs further investigation.

With the gene expression profiles of single cells from tumors, we were able to identify metabolic features that distinguish subpopulations of immune and stromal cells. This approach has the advantage of providing a direct snapshot of the metabolic landscape of tumors and their microenvironment consisting of numerous known and unknown types of cells whose metabolism is greatly influenced by the interactions between them and the shortage of nutrients in the tumor microenvironment[11,58,59]. We found that some non-malignant cell subpopulations, Ths and Tregs for instance, adopt metabolic phenotypes distinct from what they show in ex vivo culture conditions[38]. Metabolic reprogramming of CAFs compared to myofibroblasts was also shown to involve more pathways than what is currently known. These results highlight the great impact of the tumor microenvironment on cellular metabolism. It is worth noting that currently the ability to characterize metabolic phenotypes of cell subpopulations is still limited by the number of single cells that can be profiled at the same time due to the diversity of cell types and noisy gene expression in single cells. Improvement in single-cell omics techniques will help address this issue and provide higher resolution in identifying cell subpopulations with different metabolic phenotypes.

To summarize, this study offers a global picture of metabolic gene expression in single tumor and non-tumor cells from the highly complex tumor microenvironment. These cells display metabolic activities distinct from the average pattern at the bulk level. Although this study only focused on two tumor types consisting of the highest quality data at this time that allowed for the current scope of analysis, the principles about metabolic landscape of single cells in tumors – the metabolic plasticity and activity of malignant cells, the dominant role of mitochondrial programs in shaping metabolic heterogeneity of malignant and non-malignant cells, and the metabolic features of immune cell subtypes - were applicable to both tumor types, and the data analysis pipeline that we developed here can easily be extended to datasets of other tumor types. With the rapid development of novel single-cell omics techniques and accumulation of data in more tumor types and patients, we are optimistic that a comprehensive portrait of metabolic features of every unique tumor cell will emerge in the near future.

## Methods

**Data processing.** Gene expression levels were quantified using the metric log2 (TPM + 1). Tumors and non-malignant cell types containing <50 cells were excluded from the downstream analysis. Missing gene expression values were imputed using the scImpute algorithm[60] with default parameters and TPM values and gene lengths (for a gene associated with multiple transcripts, the length of the longest transcript was used) as the input. Imputation was only applied to genes with dropout rates (i.e. the fraction of cells in which the corresponding gene has

zero expression value) larger than 50% to avoid over-imputation[60]. Lists of metabolic genes and pathways were obtained from the KEGG database (http://www.kegg.jp). The imputed expression values were then used in clustering analysis using the t-SNE method[26] implemented in the Rtsne package[61] with default parameters. Bulk RNA-seq data for matched HNSCC tumor and normal samples were downloaded from TCGA database (https://portal.gdc.cancer.gov/).

**Evaluation of normalization methods.** The relative log expression (RLE) method[62] was implemented using the estimateSizeFactorsForMatrix function in DESeq2[63]. The trimmed mean of M-values (TMM)[64] and upper quartile[65] methods were performed using the calcNormFactors function in the edgeR package[64]. For deconvolution normalization[27] for scRNA-seq data with annotated cell type information, the computeSumFactors function in the scran package was used to compute cell type-specific size factors[66]. TPM values were transformed to read counts by multiplying TPM values and gene lengths (for genes with multiple transcripts, the length of the longest transcript was used). Normalized gene counts were computed by dividing read counts by the size factor corresponding to the cell and then transforming back to TPM by dividing the gene length. To avoid noise caused by low expressed and undetected genes, only genes with dropout rate <0.75 (i.e. genes with non-zero expression levels in at least 25% of the cells) were used as the reference genes to do normalization. The distributions of relative gene expression values (defined in the Calculation of pathway activity section) in different cell types were used to evaluate the performances of these methods. The method minimizing differences in distributions of relative gene expression levels between cell types was selected for the following analysis.

**Calculation of pathway activity.** For the $i$-th metabolic gene, we first calculated its mean expression level across cells of the $j$-th cell type:

$$E_{i,j} = \frac{\sum_{k=1}^{n_j} g_{i,k}}{n_j}, \ i \in 1 \ldots M, j \in 1 \ldots N \quad (1)$$

In which $n_j$ is the number of cells in the $j$-th cell type, $g_{i,k}$ is the expression level of the $i$-th gene in the $k$-th cell in this cell type, $M$ is the number of metabolic genes, and $N$ is the number of cell types. The relative expression level of the $i$-th gene in the $j$-th cell type was then defined as the ratio of $E_{i,j}$ to its average over all cell types:

$$r_{i,j} = \frac{E_{i,j}}{\frac{1}{N} \sum_j^N E_{i,j}} \quad (2)$$

Here $r_{i,j}$ quantifies the relative expression level of gene $i$ in cell type $j$ comparing to the average expression level of this gene in all cell types. A $r_{i,j}$ value >1 means that expression level of gene $i$ is higher in cell type $j$ compared to its average expression level over all cell types. The pathway activity score for the $t$-th pathway and the $j$-th cell type was then defined as the weighted average of $r_{i,j}$ over all genes included in this pathway:

$$p_{t,j} = \frac{\sum_{i=1}^{m_t} w_i \times r_{i,j}}{\sum_{i=1}^{m_t} w_i} \quad (3)$$

Where $p_{t,j}$ represents the activity of the $t$-th pathway in the $j$-th cell type, $m_t$ is the number of genes in the pathway $t$, $w_i$ is the weighting factor equal to the reciprocal of number of pathways that include the $i$-th gene. To avoid the possibility that pathway activity scores were affected by genes with low expression level or high drop-out rates, we excluded the outliers in each pathway defined by genes with relative expression levels greater than three times 75th percentile or below 1/3 times 25th percentile. Statistical significance of higher or lower pathway activity in a specific cell type was then evaluated by a random permutation test, in which the cell type labels were randomly shuffled for 5000 (for the scRNA datasets) or 1000 times (for the TCGA data) to simulate a null distribution of the pathway activity scores and compare to the pathway activity scores in the original, non-shuffled dataset. For the pathway activity score $p_{t,j}$, we then calculated a p-value defined as the fraction of random pathway activity scores larger than $p_{t,j}$ (if $p_{t,j}$ is >1) or smaller than $p_{t,j}$ (if $p_{t,j}$ is <1) to assess if activity of this pathway is significantly higher or lower in this cell type than average.

**Analyzing heterogeneity of metabolic pathways.** The PCA analysis was applied on the log2-transformed TPM (log2(TPM + 1)) values without imputation of missing values. The function prcomp in R was used to perform the PCA analysis. For each metabolic gene, we computed its PCA score defined as the sum of absolute values of the loadings of this gene in the top PCs that in total account for at least 80% of the variance to measure variability of gene expression across cells. We then sorted the PCA scores in descending order and applied GSEA analysis to the ranked list of genes to identify metabolic pathways enriched in genes with highest variability. GSEA analysis was done by the software javaGSEA available at http://software.broadinstitute.org/gsea/downloads.jsp with the option pre-ranked and default parameters. The hypoxia signature genes were retrieved from the gene set HALLMARK_HYPOXIA in the molecular signature database (MSigDB) available at http://software.broadinstitute.org/gsea/msigdb/index.jsp.

**Analysis of non-malignant cell subtypes.** T cells were classified as CD4+ or CD8+ based on expression levels of *CD4* and *CD8A*. T cells with *CD4* expression level higher than 1 and *CD8A* expression level lower than 1 were classified as CD4+ T cells, while those with *CD4* expression level lower than 1 and *CD8A* expression level higher than 1 were classified as CD8+ T cells. Cells with *CD4* and *CD8A* expression levels both higher than 1 were excluded from the following analysis. CD4+ T cells with the total expression level of *FOX3P* and *CD25* higher than 2 were further defined as Tregs, while CD4+ T cells without these two genes expressed (i.e. both genes have zero expression values in these cells) were defined as Ths. For fibroblast cells, after excluding cells with *FOS* and *VIM* expression levels both <1, k-means clustering analysis was performed on the expression levels of a set of gene markers (Fig. 5f) to classify them into CAFs and myofibroblasts. The metabolic gene expression profiles were then compared between different cell subtypes using GSEA with the following parameters: nperm = 1000, metric = Diff_of_Classes, permute = gene_set, set_max = 500, set_min = 5. The metabolic pathways with GSEA nominal $p$-value < 0.05 were considered as significant.

**Reporting summary.** Further information on research design is available in the Nature Research Reporting Summary linked to this article.

## Data availability
Processed gene expression profiles for melanoma and HNSCC were retrieved from Gene Expression Omnibus (GEO; https://www.ncbi.nlm.nih.gov/gds) under accession numbers GSE72056 and GSE103322. Raw and processed gene expression values and annotation of cell and tumor types used in this study are available at https://doi.org/10.6084/m9.figshare.7174922 (ref. [67]). All other data supporting the findings of this study are available within the article and its Supplementary Information files.

## Code availability
Computer codes used in this study are available at the GitHub page of the Locasale Lab: https://github.com/LocasaleLab/Single-Cell-Metabolic-Landscape.

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

## Acknowledgements

We thank members of the Locasale laboratory for helpful discussions. Z.D. thanks Annamarie Allen for help with the text. Support from National Institutes of Health (R01CA193256, R00CA168997), the American Cancer Society (RSG-16-214-01-TBE), and support from the Duke Compute Cluster and Data Commons Storage for computational resources are gratefully acknowledged.

## Author contributions

Z.X., Z.D., and J.W.L. designed the study and wrote the manuscript. Z.X. performed the data analysis with help from Z.D. All authors have read and approved the final manuscript.

## Additional information

**Competing interests:** The authors declare no competing interests.

