## [Peer Review File · Nature Communications]

Reviewers' Comments:

Reviewer #1:

Remarks to the Author:

In the manuscript by Xiao, Dai, and Locasale, the authors describe the analysis of single cell RNA sequencing data from the perspective of metabolic genes. They use the enhanced resolution of single cell data to tease apart the contribution of variation in metabolic gene expression to tumor-specific gene expression changes. The authors report that the majority of the variation in the data is associated with mitochondrial metabolic gene expression. They go on to analyze metabolic gene expression programs of specific malignant and immune cell types. While the analysis is novel, several aspects of it are confusing, and more importantly it appears that few specific conclusions are drawn from the results.

Major Comments:

1. I understand the premise that it is inherently interesting to study metabolic genes, because inferences could potentially be made on metabolic pathway activity from gene expression data. However, what is unclear to me is whether there is anything inherently "special" about metabolic genes. For example, in Fig 1d/e, the authors show that malignant cells cluster according to tumor when running tSNE on metabolic genes. Is this property of clustering within tumor-of-origin unique to metabolic genes, or would a randomly selected set of genes also exhibit this pattern?
2. Can the authors explain why the results of Figure 2a/b appear so imbalanced? I was surprised to observe that malignant cells generically appear to upregulate a large variety of metabolic pathways, but downregulate comparatively few (with the opposite trend of a vast downregulation of metabolic pathways in the other cell types). Is this a real effect, or an artifact? Is it associated with the library size/number of reads of each cell? It may be useful in this scenario to repeat the analysis on non-metabolic pathways. If the same phenomenon is observed across all pathways, it would suggest the effect is artifactual.
3. Related to the above comments, is it possible to ascribe some sort of p-value/measure of statistical significance to the pathway scores in Figure 2a/b?
4. Related to the comment above, assuming the reported pathway changes are not artifactual, what are we to conclude from them? The authors write that the malignant cells undergo a "global up-regulation of metabolic activity." I understand that it is not feasible to measure flux at this kind of resolution, but it seems suspect to conclude that simply because gene expression levels are higher, that the corresponding flux through a metabolic pathway is similarly higher. This is a tough comment to respond to, and I don't expect the authors to be able to address it head-on, but I think a careful and clear clarification of the conclusions of this analysis in the text is called for. If we cannot infer changes in flux, what is it that we should conclude from these findings?
5. I found the analysis of metabolic variation intriguing, but the methodological approach puzzling. If the question is to understand which metabolic genes contribute the greatest amount to the variation in the data, why not simply calculate this quantity directly using the gene expression data, e.g. by calculating the coefficient of variation on the log(transcripts per million) of each gene? Summing the absolute values of the loadings in the first few principal components is an alternative approach, but it feels both (1) indirect, and (2) inappropriate because PCA inherently is looking for linear combinations of (rather than individual) metabolites which maximize variance. Does running GSEA using a metric like the coefficient of variation produce similar results? Related to this, I would also ask the authors to confirm that their results are not corrupted by total average expression (i.e. the sum of TPMs) of the constituent genes in their geneset. OXPHOS is itself among the most highly expressed sets of genes in many cell types, and if this is not properly accounted for, it could appear that it is driving the majority of variation in gene expression simply because it is so highly expressed.

6. Given that the expression of genes involved in glycolysis and OXPHOS is correlated to hypoxia, could the authors describe the gene expression signature of OXPHOS low/glycolysis low/hypoxia low cells?

7. The analysis of metabolic programs in immune cells was again novel, but also felt lacking a conclusion. Is there any functional data, either already-published or newly produced by the authors, which can help to validate and give additional context to the findings reported in Figure 5b/c/d/e? Do these pathways truly carry different levels of flux in different cell types? If this is not possible, is there any additional data on metabolite levels in these cell types which would offer (albeit indirect) supportive evidence of distinct metabolic activity in OXPHOS and other pathways across CD8+/CD4+ or Th/Treg cell types?

8. Despite many of my critical comments above, I would also like to commend the authors. They have organized, annotated, and released their code on Github, which is valuable to many others who will seek to apply reproduce and build on their approach in other datasets.

Reviewer #2:

Remarks to the Author:

Xiao et al investigate previously published single-cell RNA-seq data aiming to better understand the heterogeneity of tumour metabolism, taking into account both the cancer and normal cell compartments. This work is timely complement to the original reports of the single-cell RNA-seq measurements, which were based on a global analysis of the gene expression profiles leaving open the investigation of the tumour metabolism. Using their computational analysis Xia et al uncover some interesting features about the metabolism of cancer cells and non-cancerous cells in human tumours that cannot be deduced from the bulk gene expression profiles of tumours. The observation that, at the single cell level, the gene signatures of hypoxia positively correlate with those of oxidative phosphorylation is interesting and it should be motivation for future in vitro or in vivo experimental work. There are however a number of points where further work is required to support the conclusions made.

1- Section starting at line 79. Since the metabolic analysis is based on a subset of the genes, it is important to understand up to what extent the expression profiles associated with metabolic gene set are statistically similar or different to the expression profiles from the whole set of genes. Specifically, how the clustering in Fig. 1d-g compares to the same clustering using as input the expression profiles of all genes? The comparison could be done using relative mutual information $I(A,B)/\max\{I(A,B)\}$, where A and B are the two different clusterings, $I(A,B)$ is the observed mutual information between the two clusterings and $\max\{I(A,B)\}$ its expected maximum.

2- Line 162. At this point the authors investigate differences between observations made using single cell vs bulk gene expression profiles. As a reference the use bulk gene expression profiles from the TCGA. This is not a proper design to make that test because we cannot warranty that the TCGA samples are statistically equivalent to the bulk expression profiles of the samples for which single cell data is available. Although there is no bulk data for the latter, the authors can computationally reconstruct the bulk gene expression profiles by adding the single cell gene expression profiles associated with each tumour, weighting each cell compartment by the corresponding fraction of the total tumour mass. Then the authors should compare the analysis made on the single cell data with that made on the reconstructed bulk gene expression profiles. In this way the authors will eliminate any confounding factors due to that analysis of different tumour samples and different experimental protocols.

3- Line 210. The authors report: "To our surprise, we also found that OXPHOS significantly correlated with glycolysis". I understand there is a common credo that OXPHOS and glycolysis antagonize each other, but that that is mostly based on assumptions from propagation of ideas in

the "old" literature. The authors should provide a level of reference to evaluate the significance of that surprise. For example, the authors could repeat the same analysis using gene expression profiles for a panel of cancer cell lines cultured in vitro (e.g., Cancer Cell Line Encyclopedia). If in the latter the OXPPOS gene signature is not correlated to the glycolysis gene signature than we can say that the correlation is linked to the tumour microenvironment.

4- The overall impression from this investigation is that OXPPOS and glycolysis are the major drivers of the inter- and intra- metabolic heterogeneity of tumours, both in malignant and non-malignant cells within the tumour. This should be stressed in a summary Figure. I note that information is somehow depicted in Fig. 2a,b, but there is too much distraction in that figure. For example, the data represented in the glycolysis and OXPPOS lanes in Fig. 2a,b could be plotted using a box plot format. That would give a better idea about the magnitude of the differences reported, from the malignant cells to the non-malignant cells and between non-malignant cells. This way of plotting the data will also allow for a better sense of up to what extent some of the reported changes are trends or actually significant. For example, in line 283 it is reported that glycolysis is upregulated in CAFs relative to myofibroblast with a significance of 0.048. We need to see the boxplot with outliers highlighted as scatter to get a better idea of whether that is significant.

Responses to both reviewers:

We thank the reviewers for their helpful comments and constructive suggestions and the positive evaluation of our work. When additional analysis was conducted, the data are presented both in the revised manuscript and also in response figures shown below. Modifications to the text are shown in yellow font.

Since both reviewers have raised the comment regarding the difference between metabolic genes and non-metabolic genes in generating the clustering patterns of the malignant and non-malignant cells, we have repeated the t-SNE analysis in (1) a randomly selected set of genes (Response Figure 1a-d); (2) the complete set of genes included in the single-cell RNA-seq datasets (Response Figure 1e-h), and compared the resulting clustering patterns with that generated using the metabolic genes using the relative mutual information metric (Response Table 1). We found that for both melanoma and HNSCC, clustering patterns of the random gene set and the complete gene set are both highly consistent with that generated with metabolic genes (relative mutual information close to 1), in which malignant cells were clustered according to their tumor-of-origin while non-malignant cells were clustered by cell type.

Response Figure 1 (now Supplementary Figure 3) t-SNE plots generated using different gene sets. (a) t-SNE plot of expression profiles of randomly selected genes in malignant cells from the melanoma dataset. The color of each dot indicates the tumor from which the cell was derived. (b) same as in (a) but for non-malignant cells. (c) same as in (a) but for the HNSCC dataset. (d) same as in (b) but for the HNSCC dataset. (e) same as in (a) but for the complete gene set. (f) same as in (b) but for the complete gene set. (g) same as in (c) but for the complete gene set. (h) same as in (d) but for the complete gene set.

Taken together, these results suggest that the distinct clustering patterns of malignant cells and non-malignant cells reflect the relationship between cell phenotypes and the tumor microenvironment. Gene expression programs in malignant cells are in general more flexible and responsive to alterations in genetic and environmental factors that affect both metabolic and non-metabolic pathways. We have included the additional analyses and relevant discussions in the revised manuscript.

Response Table 1 (now Supplementary Table 1). Relative mutual information between t-SNE results generated using different gene sets

Data set		Metabolic gene set	Metabolic gene set
		vs Complete gene set	vs Random gene set
Melanoma	Malignant cells	1	1
	Non-malignant cells	0.96	0.96
HNSCC	Malignant cells	1	1
	Non-malignant cells	1	1

Reviewer #1 (Remarks to the Author):

In the manuscript by Xiao, Dai, and Locasale, the authors describe the analysis of single cell RNA sequencing data from the perspective of metabolic genes. They use the enhanced resolution of single cell data to tease apart the contribution of variation in metabolic gene expression to tumor-specific gene expression changes. The authors report that the majority of the variation in the data is associated with mitochondrial metabolic gene expression. They go on to analyze metabolic gene expression programs of specific malignant and immune cell types. While the analysis is novel, several aspects of it are confusing, and more importantly it appears that few specific conclusions are drawn from the results.

We thank the reviewer for the positive remarks and constructive suggestions. We have included additional analysis and clarification in the revised manuscript to address the reviewer's concerns.

Major Comments:

1. I understand the premise that it is inherently interesting to study metabolic genes, because inferences could potentially be made on metabolic pathway activity from gene expression data. However, what is unclear to me is whether there is anything inherently "special" about metabolic genes. For example, in Fig 1d/e, the authors show that malignant cells cluster according to tumor when running tSNE on metabolic genes. Is this property of clustering within tumor-of-origin unique to metabolic genes, or would a randomly selected set of genes also exhibit this pattern?

We thank the reviewer for raising this comment. We completely agree that it is important to compare the single-cell transcriptomic landscape between metabolic genes and non-metabolic genes. Since another reviewer has also raised this concern, we have responded to it in the section 'Responses to both reviewers' at the beginning of the response letter. Please refer to our general responses for more details.

2. Can the authors explain why the results of Figure 2a/b appear so imbalanced? I was surprised to observe that malignant cells generically appear to upregulate a large variety of metabolic pathways, but downregulate comparatively few (with the opposite trend of a vast downregulation of metabolic pathways in the other cell types). Is this a real effect, or an artifact? Is it associated with the library size/number of reads of each cell? It may be useful in this scenario to repeat the analysis on non-metabolic pathways. If the same phenomenon is observed across all pathways, it would suggest the effect is artifactual.

We thank the reviewer for raising this concern. In the previous version of manuscript, we had made attempts to avoid artifacts by applying appropriate data normalization method that resulted in similar distributions of relative gene expression across cell types (deconvolution method, Supplementary

Figure 4). After the data normalization, all cell types show similar extent of up- and down-regulation of gene expression. Thus, we believe that the imbalanced distribution of pathway score for metabolic pathways in malignant cells indicates global up-regulation of metabolic pathways.

To further clarify this point and exclude the possibility of artifacts, we have now extended the analysis of pathway scores to non-metabolic pathways (Response Figure 2). We found that the distributions of pathway scores in non-metabolic pathways were similar among all cell types with median values close to 1 (i.e. no global up-regulation or down-regulation of pathway activities in any cell type), indicating that the global up-regulation in malignant cells is a unique feature of metabolic pathways. We have included these results as Supplementary Figure 6 in the revised manuscript.

Moreover, we agree with the reviewer that sequencing depths and library sizes are important factors to consider to fully exclude artifacts. However, these variables are currently not available. The gene expression levels in these two datasets are quantified using TPM values in which the variation in sequencing depth has already been corrected for. It is also worth noting that for single cell RNA-seq data, the relative gene expression level and pathway score could be greatly affected by genes with low expression level or high drop-out rates, thereby biasing the pathway scores towards values much higher than 1. Thus, in the revised manuscript, we have also re-evaluated the pathway scores after including an additional step of removing outliers defined by genes whose relative expression levels are higher than 3 times 75th percentile or below 1/3 times 25th percentile in each pathway. The resulting metabolic pathway activities still exhibit the pattern of global up-regulation in malignant cells. The updated results are included in the revised manuscript (Figure 2).

3. Related to the above comments, is it possible to ascribe some sort of p-value/measure of statistical significance to the pathway scores in Figure 2a/b?

We thank the reviewer for this suggestion and apologize for the confusion. In the original manuscript, we have already evaluated the statistical significance of pathway scores using a random permutation test, in which we randomly shuffled the cell type labels to generate a null distribution of pathway scores (details in the section ‘Calculation of pathway activity’). Only pathway scores with statistical significance (p -value<0.05) were shown in the heatmaps in Figure 2a and b. In the revised manuscript, we have included additional clarification of this point and reported the pathway scores and associated p -values as a supplementary file (Supplementary Table 2).

4. Related to the comment above, assuming the reported pathway changes are not artifactual, what are we to conclude from them? The authors write that the malignant cells undergo a "global up-regulation of metabolic activity." I understand that it is not feasible to measure flux at this kind of resolution, but it seems suspect to conclude that simply because gene expression levels are higher, that the corresponding flux through a metabolic pathway is similarly higher. This is a tough comment to respond to, and I don't expect the authors to be able to address it head-on, but I think a careful and clear clarification of the conclusions of this analysis in the text is called for. If we cannot infer changes in flux, what is it that we should conclude from these findings?

We thank the reviewer for the helpful suggestion and thoughtful remarks. We completely agree with the reviewer that metabolic gene expression levels and metabolic fluxes are not equivalent. The key point here is to what extent changes in metabolic gene expression reflect changes in metabolic fluxes. Although measurements of metabolic fluxes at the single-cell level are currently not available due to technical limitations, evidence from bulk measurements of metabolic gene expression, metabolic flux and metabolite abundance has suggested positive correlation between expression levels of metabolic genes, metabolic fluxes and abundance of metabolites (e.g. Mehrmohamadi et al, Cell Reports 2014, PMID: 25456139; Peng et al, Cell Reports 2018, PMID: 29617665). Thus, we believe that the metabolic gene expression features we identified in this study indicate overall trends of metabolic activities in single malignant and non-malignant cells. We have included additional discussion about this point in the revised manuscript.

5. I found the analysis of metabolic variation intriguing, but the methodological approach puzzling. If the question is to understand which metabolic genes contribute the greatest amount to the variation in the data, why not simply calculate this quantity directly using the gene expression data, e.g. by calculating the coefficient of variation on the log(transcripts per million) of each gene? Summing the absolute values of the loadings in the first few principal components is an alternative approach, but it feels both (1) indirect, and (2) inappropriate because PCA inherently is looking for linear combinations of (rather than individual) metabolites which maximize variance. Does running GSEA using a metric like the coefficient of variation produce similar results? Related to this, I would also ask the authors to confirm that their results are not corrupted by total average expression (i.e. the sum of TPMs) of the constituent genes in their geneset. OXPHOS is itself among the most highly expressed sets of genes in many cell types, and if this is not properly accounted for, it could appear that it is driving the majority of variation in gene expression simply because it is so highly expressed.

We thank the reviewer for raising this concern.

We used PCA loadings to identify the highly variable genes and control for potential confounders such as baseline expression levels and noise in gene expression. A strategy similar to our PCA-based approach has also been applied to select highly variable genes and identify cell subpopulations based on single cell RNA-seq datasets in other studies (e.g. Fan et al. Nature Methods 2016, PMID: 26780092). Coefficient of variation (CV), on the other hand, tends to be higher for genes with lower expression levels. To clarify this point, we have evaluated the correlation between baseline gene expression levels and heterogeneity metrics including PCA score, CV and standard deviation (SD)

(used to quantify heterogeneity in gene expression for example in Elham et al. Cell 2018, PMID: 29961579). We found that CV monotonically decreased with increasing expression level, meaning that genes with higher CV tend to have lower expression levels (Response Figure 3a). On the other hand, correlations between average expression and SD (Response Figure 3b) or PCA score (Response Figure 3c) were lower, and genes with the highest PCA scores or SD values were neither these with highest expression levels nor those with lowest expression. Thus, we believe that the PCA score and SD are better metrics for heterogeneity in gene expression since they are less sensitive to confounding by changes in average gene expression level.

Response Figure 3 (now supplementary Figure 9) Comparison of PCA score, CV and SD for quantifying intratumoral heterogeneity. (a) Scatter plot comparing mean gene expression and coefficient of variation (CV) in malignant cells from the melanoma tumor MEL3. (b) Same as in (a) but for standard deviation (SD). (c) Same as in (a) but for PCA score. (d) Metabolic pathways enriched in genes with highest CV in melanoma dataset. (e) Same as in (d) but for genes with highest SD. (f) Same as in (d) but for genes with highest PCA score. (g) Swarm plots comparing distributions of gene expression levels in OXPHOS and steroid hormone biosynthesis.

We have also repeated the GSEA analysis for highly heterogenous genes identified using CV and SD (Response Figure 3d,e) and found that the enriched pathways were highly consistent between the PCA

score (Response Figure 3f) and SD, while steroid hormone biosynthesis, the top pathway associated with high CV, had much lower gene expression levels and visibly less variation in gene expression compared to OXPPOS, the top heterogenous pathway identified by SD and the PCA score (Response Figure 3g). We have added these results to the revised manuscript as Supplementary Figure 9.

6. Given that the expression of genes involved in glycolysis and OXPPOS is correlated to hypoxia, could the authors describe the gene expression signature of OXPPOS low/glycolysis low/hypoxia low cells?

We thank the reviewer for this helpful suggestion. We have included additional analysis to identify the gene expression signatures of OXPPOS low/glycolysis low/hypoxia low cells (Response Figure 4) in the revised manuscript. Briefly, we identified genes differentially expressed between the two groups of cells with the highest or lowest expression levels of genes in OXPPOS, glycolysis and hypoxia and applied GO enrichment analysis to identify biological functions enriched in genes up-regulated in cells with lowest expression levels of genes in OXPPOS, glycolysis and hypoxia pathways. We found that in both melanoma and HNSCC, genes up-regulated in the OXPPOS low/glycolysis low/hypoxia low cells were related to the GO term ‘negative regulation of execution phase of apoptosis’, suggesting that this subpopulation of cells is associated with reduced apoptosis which may facilitate cancer progression. These results are included in the revised manuscript as Figure 3g, h and Supplementary Figure 11.

7. The analysis of metabolic programs in immune cells was again novel, but also felt lacking a conclusion. Is there any functional data, either already-published or newly produced by the authors, which can help to validate and give additional context to the findings reported in Figure 5b/c/d/e? Do these pathways truly carry different levels of flux in different cell types? If this is not possible, is there any additional data on metabolite levels in these cell types which would offer (albeit indirect) supportive evidence of distinct metabolic activity in OXPPOS and other pathways across CD8+/CD4+ or Th/Treg cell types?

We thank the reviewer for raising this concern. First, as we noted previously, although measurements of metabolic fluxes at the single-cell level are currently unavailable, there is evidence supporting a positive correlation between metabolic gene expression levels and metabolic fluxes. Thus, we expect that the directions of changes we observed with metabolic gene expression levels are consistent with

that of metabolic fluxes. Moreover, there is also evidence from experimental studies that validates our computational analysis. For instance, there are studies showing that mitochondrial metabolism is up-regulated in Tregs compared to Ths (e.g. Weinberg et al, Nature 2019, PMID: 30626970; Angelin et al. Cell Metabolism 2017, PMID: 28416194), which is consistent with our analysis. We have included additional references and discussion in the revised manuscript.

8. Despite many of my critical comments above, I would also like to commend the authors. They have organized, annotated, and released their code on Github, which is valuable to many others who will seek to apply reproduce and build on their approach in other datasets.

We thank the reviewers for the positive comment. We are delighted to see that our code is helpful to others in the field.

Reviewer #2 (Remarks to the Author):

Xiao et al investigate previously published single-cell RNA-seq data aiming to better understand the heterogeneity of tumour metabolism, taking into account both the cancer and normal cell compartments. This work is timely complement to the original reports of the single-cell RNA-seq measurements, which were based on a global analysis of the gene expression profiles leaving open the investigation of the tumour metabolism. Using their computational analysis Xiao et al uncover some interesting features about the metabolism of cancer cells and non-cancerous cells in human tumours that cannot be deduced from the bulk gene expression profiles of tumours. The observation that, at the single cell level, the gene signatures of hypoxia positively correlate with those of oxidative phosphorylation is interesting and it should be motivation for future in vitro or in vivo experimental work. There are however a number of points where further work is required to support the conclusions made.

We thank the reviewer for the positive evaluation of our work and the constructive suggestions. We have included additional analysis as the reviewer suggested in the revised manuscript.

1- Section starting at line 79. Since the metabolic analysis is based on a subset of the genes, it is important to understand up to what extent the expression profiles associated with metabolic gene set are statistically similar or different to the expression profiles from the whole set of genes. Specifically, how the clustering in Fig. 1d-g compares to the same clustering using as input the expression profiles of all genes? The comparison could be done using relative mutual information $I(A,B)/\max\{I(A,B)\}$, where A and B are the two different clustering, $I(A,B)$ is the observed mutual information between the two clusterings and $\max\{I(A,B)\}$ its expected maximum.

We thank the reviewer for this comment. In the revised manuscript, we have repeated the t-SNE and clustering analysis with the complete set of genes and set of randomly generated genes, and compared the resulting clustering patterns with that generated with metabolic genes. We have also used the relative mutual information as the reviewer suggested to quantify the similarity between clustering patterns generated using different gene sets. Since another reviewer has also raised the same comment, we have responded to it in the section 'Responses to both reviewers' at the beginning of the response letter. Please refer to that section for more details.

2- Line 162. At this point the authors investigate differences between observations made using single cell vs bulk gene expression profiles. As a reference the use bulk gene expression profiles from the TCGA. This is not a proper design to make that test because we cannot warranty that the TCGA samples are statistically equivalent to the bulk expression profiles of the samples for which single cell data is available.

Although there is no bulk data for the latter, the authors can computationally reconstruct the bulk gene expression profiles by adding the single cell gene expression profiles associated with each tumour, weighting each cell compartment by the corresponding fraction of the total tumour mass. Then the authors should compare the analysis made on the single cell data with that made on the reconstructed bulk gene expression profiles. In this way the authors will eliminate any confounding factors due to that analysis of different tumour samples and different experimental protocols.

We thank the reviewer for raising this concern. We completely agree with the reviewer that factors such as batch effect and difference in experimental procedures may result in substantial difference between the bulk samples from TCGA and the samples used in the single cell RNA-seq study. Ideally this issue could be addressed by reconstructing bulk tumor and normal samples for the single cell dataset as the reviewer suggested. However, all cells included in the single cell dataset were derived from tumors, resulting in difficulty in reconstructing bulk gene expression profiles for normal tissue, since non-malignant cells in the tumor microenvironment also undergo reprogramming of cellular metabolism thus being very different from the same types of cells in normal tissues.

Nevertheless, despite the difficulty in reconstructing bulk normal samples for the single cell dataset, we can still reconstruct gene expression profiles for the bulk tumors by pooling the gene expression profiles of single cells derived from the same tumor to confirm that they are comparable to the tumor samples in TCGA. In the revised manuscript, we have reconstructed bulk tumor gene expression profile for each patient using all single malignant and non-malignant cells derived from this patient to evaluate the similarity between TCGA and the single cell dataset (Response Figure 5). We have also reconstructed bulk T cell gene expression profile by pooling randomly selected single T cells as a negative control. Indeed there were very strong correlations between TCGA gene expression profiles and the reconstructed bulk tumor gene expression profiles, while the correlation coefficients between the reconstructed bulk T cell sample and the TCGA gene expression profiles were weaker, suggesting that the reconstructed bulk tumors, but not the pooled T cells, closely resemble the TCGA tumor samples in terms of gene expression profiles. We believe that this is a strong evidence to support that the tumor samples from TCGA and those from the single cell RNA-seq dataset are approximately equivalent. We have included these results and corresponding discussions in the revised manuscript.

3- Line 210. The authors report: “To our surprise, we also found that OXPHOS significantly correlated with glycolysis”. I understand there is a common credo that OXPHOS and glycolysis antagonize each other, but that that is mostly based on assumptions from propagation of ideas in the “old” literature. The authors should provide a level of reference to evaluate the significance of that surprise. For example, the authors could repeat the same analysis using gene expression profiles for a panel of cancer cell lines cultured in vitro (e.g., Cancer Cell Line Encyclopedia). If in the latter the OXPHOS gene signature is not

correlated to the glycolysis gene signature than we can say that the correlation is linked to the tumour microenvironment.

We thank the reviewer for this helpful suggestion. Following this reviewer's suggestion, we have performed additional analysis to evaluate the correlation between OXPHOS and glycolysis using gene expression profiles of cultured cancer cell lines from the Cancer Cell Line Encyclopedia (Response Figure 6). The analysis showed that only the correlation between glycolysis and hypoxia was preserved in the cultured cancer cells, while the correlation between gene signature of OXPHOS and glycolysis and that between OXPHOS and hypoxia were lost. We have included these results in the revised manuscript as Figure 3f and added discussion of this point.

4- The overall impression from this investigation is that OXPHOS and glycolysis are the major drivers of the inter- and intra- metabolic heterogeneity of tumours, both in malignant and non-malignant cells within the tumour. This should be stressed in a summary Figure. I note that information is somehow depicted in Fig. 2a,b, but there is too much distraction in that figure. For example, the data represented in the glycolysis and OXPHOS lanes in Fig. 2a,b could be plotted using a box plot format. That would give a better idea about the magnitude of the differences reported, from the malignant cells to the non-malignant cells and between non-malignant cells. This way of plotting the data will also allow for a better sense of up to what extent some of the reported changes are trends or actually significant. For example, in line 283 it is reported that glycolysis is upregulated in CAFs relative to myofibroblast with a significance of 0.048. We need to see the boxplot with outliers highlighted as scatter to get a better idea of whether that is significant.

We thank the reviewer for raising this point. We have revised the figures to include boxplots comparing average expression levels of glycolysis and OXPHOS genes across different cell types and included one-way ANOVA p-values to evaluate the significance level of difference in gene expression across cell types (Response Figure 8). We have also included boxplots to compare average expression levels of all pathways significantly differentially expressed between the subtypes of T cells and fibroblasts, and performed Wilcoxon's rank-sum test to confirm the significance of differential expression of the pathways between cell subtypes (Response Figure 9). Moreover, we have updated Figure 5c to correct the mistake that in the previous version, all bars in this figure were colored regardless of whether the enrichment of the corresponding pathway was significant.

Response Figure 9 (now supplementary Figure 13) Distributions of average expression level of pathways differently expressed in cell subtypes. (a) Distributions of average expression of all OXPHOS genes (left) and the leading-edge genes enriched in the GSEA analysis (right) in CD4+ and CD8+ T cells in the melanoma dataset. (b) Distributions of average gene expression levels for pathways differentially expressed between CD4+ and CD8+ T cells in the HNSCC dataset. (c) Same as in (b) but for pathways differentially expressed in Th and Treg cells in the melanoma dataset. (d) Same as in (c) but for the HNSCC dataset. (e) same as in (d) but for pathways differentially expressed in CAF cells and Myofib cells in the HNSCC dataset. P-values shown in each graph were computed using one-sided Wilcoxon's rank-sum test for comparison of gene expression levels between the cell subtypes.

Reviewers' Comments:

Reviewer #1:

Remarks to the Author:

The authors have sufficiently addressed my critiques and comments.

I would make one additional suggestion which may help clarify one of my concerns. Regarding the question of what to conclude from generic upregulation of metabolic genes (but not other sets of genes), it may be useful to think about the "compositional" nature of this data. My impression of the author's finding is that, from a compositional/proportional perspective, the tumor cells are actually dedicating more of their fixed sum of "transcriptional resources" (which may/may not translate to bona fide changes in protein levels) to metabolic genes. While this still really does not lead to a conclusion about metabolic flux, it does implicitly suggest that if *every* enzyme is at higher abundance, then the per-cell turnover rate of every reaction may be higher. These are simply my thoughts, and the authors are free to ignore them or use them as they please.

Reviewer #2:

Remarks to the Author:

As I comment in my first review this is an interesting work with some points to be addressed. The authors have addressed most of my previous comments and by doing so the manuscript has improved significantly. However, point 2 was not addressed to a full extent. This point is related to two major conclusions of this work. Paraphrasing the abstract

A- "We find that malignant cells in general have higher metabolic activity and higher metabolic variation than previously observed from studies of bulk tumor comparisons.

B- "Indeed, most of the observed metabolic variation of single tumor and normal cells were found to be inconsistent with comparisons with bulk tumor samples."

If these conclusions are indeed true then we need to invest more resources into conducting single-cell analyses to obtain the correct understanding of tumor metabolism. My point 2 was that the analysis provided by the authors is not sufficient to support those conclusions. Their analysis was potentially flawed because the cohorts for the bulk and single cell data are different. I suggested they could address that caveat by reconstructing the bulk gene expression from the single cell gene expression profiles.

In their response the authors reconstructed the bulk gene expression data and went on to show that it is highly correlated with the TCGA bulk gene expression data. However, I do not understand why they stopped there. It is straightforward for the authors to conduct, I have previously requested, the metabolic pathway analysis using the reconstructed gene expression data. Specifically, what I request is a statistical test for the conclusions A and B cited above, using as input the single cell and the reconstructed bulk expression profiles. Since this has not been provided I still consider this work unsuitable for publication.

The authors cited as a limitation the lack of representation of tumor stroma cells in their single cell data. However, this potential limitation appears to be not relevant when they compared the reconstructed bulk gene expression data with the TCGA data. That evidence can be taken as support of the validity of the reconstructed bulk gene expression data, which can then be used to carry on my request.

From my end point the request stands. The authors have not provided a proper statistical test to claim A and B above. Either they address that point using a reconstructed bulk gene expression

profile or they would require to design a new study where the test could be conducted using bulk and single cell data fro the same samples.

Reviewer #1 (Remarks to the Author):

The authors have sufficiently addressed my critiques and comments.

I would make one additional suggestion which may help clarify one of my concerns. Regarding the question of what to conclude from generic upregulation of metabolic genes (but not other sets of genes), it may be useful to think about the "compositional" nature of this data. My impression of the author's finding is that, from a compositional/proportional perspective, the tumor cells are actually dedicating more of their fixed sum of "transcriptional resources" (which may/may not translate to bona fide changes in protein levels) to metabolic genes. While this still really does not lead to a conclusion about metabolic flux, it does implicitly suggest that if **every** enzyme is at higher abundance, then the per-cell turnover rate of every reaction may be higher. These are simply my thoughts, and the authors are free to ignore them or use them as they please.

We thank the reviewer for the positive remarks and insightful thoughts. In the revised manuscript, we have included more discussion of how the global up-regulation of metabolic genes in single malignant cells is related to changes in metabolic fluxes in these cells.

Reviewer #2 (Remarks to the Author):

As I comment in my first review this is an interesting work with some points to be addressed. The authors have addressed most of my previous comments and by doing so the manuscript has improved significantly.

We greatly appreciate the reviewer's positive evaluation of the revisions. We have performed additional analysis to address the remaining concern raised by this reviewer.

However, point 2 was not addressed to a full extent. This point is related to two major conclusions of this work. Paraphrasing the abstract

A- "We find that malignant cells in general have higher metabolic activity and higher metabolic variation than previously observed from studies of bulk tumor comparisons.

B- "Indeed, most of the observed metabolic variation of single tumor and normal cells were found to be inconsistent with comparisons with bulk tumor samples."

If these conclusions are indeed true then we need to invest more resources into conducting single-cell analyses to obtain the correct understanding of tumor metabolism. My point 2 was that the analysis provided by the authors is not sufficient to support those conclusions. Their analysis was potentially flawed because the cohorts for the bulk and single cell data are different. I suggested they could address that caveat by reconstructing the bulk gene expression from the single cell gene expression profiles.

In their response the authors reconstructed the bulk gene expression data and went on to show that it is highly correlated with the TCGA bulk gene expression data. However, I do not understand why they stopped there. It is straightforward for the authors to conduct, I have previously requested, the metabolic pathway analysis using the reconstructed gene expression data. Specifically, what I request is a statistical test for the conclusions A and B cited above, using as input the single cell and the reconstructed bulk expression profiles. Since this has not been provided I still consider this work

unsuitable for publication.

The authors cited as a limitation the lack of representation of tumor stroma cells in their single cell data. However, this potential limitation appears to be not relevant when they compared the reconstructed bulk gene expression data with the TCGA data. That evidence can be taken as support of the validity of the reconstructed bulk gene expression data, which can then be used to carry on my request.

From my end point the request stands. The authors have not provided a proper statistical test to claim A and B above. Either they address that point using a reconstructed bulk gene expression profile or they would require to design a new study where the test could be conducted using bulk and single cell data for the same samples.

We thank the reviewer for clarifying this point and apologize for having misunderstood it in the previous round of revision. Following this reviewer's suggestion, we have now repeated the analysis of metabolic pathway activity by directly comparing the single malignant cells with the reconstructed bulk tumors. In addition to HNSCC for which we have reconstructed bulk gene expression profiles in the first round of revision, we have also reconstructed bulk gene expression profiles for melanoma to provide more evidence to support our conclusions.

We found that for both melanoma and HNSCC, single malignant cells showed higher metabolic pathway activity (Response Figure 1a, one-sided Wilcoxon's rank-sum test p -value = $2.5e-5$ for melanoma and $4.6e-4$ for HNSCC) and higher variation in metabolic pathway activity (Response Figure 1a, standard deviation = 0.28 for single malignant cells compared to 0.16 for reconstructed bulk tumors in the melanoma dataset and 0.6 for single malignant cells compared to 0.24 for reconstructed bulk tumors in the HNSCC dataset) compared to reconstructed bulk tumors. These results serve as additional evidence to support the conclusion that single malignant cells show higher metabolic activity and variation than bulk tumors (conclusion A).

The inconsistency between single malignant cells and reconstructed bulk tumors in metabolic pathway activity was further illustrated by quantile-quantile plots comparing their distributions of metabolic pathway activity scores (Response Figure 1b) and evaluated using Kolmogorov-Smirnov test (K-S test p -value = $2.9e-5$ for melanoma and $8.7e-5$ for HNSCC). These results support the conclusion that metabolic features of single malignant cells are distinct from those of bulk tumors (conclusion B).

Taken together, these additional results are consistent with the comparison between single malignant cells and the TCGA bulk tumors, thus providing further evidence to support our key conclusions. We have included these results in the revised manuscript as Supplementary Figure 8.

Reviewers' Comments:

Reviewer #2:

Remarks to the Author:

The authors have provided a satisfactory response to my previous comments. The authors have provided evidence that the single cell data has more variation in the metabolic genes expression than the bulk data reconstructed from single cells.

Response to reviewer #2

Reviewer #2 (Remarks to the Author):

The authors have provided a satisfactory response to my previous comments. The authors have provided evidence that the single cell data has more variation in the metabolic genes expression than the bulk data reconstructed from single cells.

We thank the reviewer for the positive comments.

We have reformatted the mathematical terms throughout the manuscript to ensure that they are consistent with the editorial guidelines.

* Wherever p-values are stated in the text and figure legends, please also state the name of the statistical test.

We have included statements about the statistical tests in the text and figure legends wherever p-values are stated.

METHODS AND DATA